# Correlates of never testing for HIV among men who have sex with men in Nepal

Kiran Paudel[1,2], Kamal Gautam[2], Anjila Pandey[3], Prashamsa Bhandari[4], Manisha Dhakal[5], Md. Safaet Hossain Sujan[2], Jefferey A. Wickersham[6], Roman Shrestha[1,2,6]*

1 Nepal Health Frontiers, Tokha-5, Kathmandu, Nepal, 2 Department of Allied Health Sciences, University of Connecticut, Storrs, Connecticut, United States of America, 3 Manmohan Memorial Institute of Health Sciences, Tribhuvan University, Kathmandu, Nepal, 4 Institute of Medicine, Tribhuvan University, Kathmandu, Nepal, 5 Blue Diamond Society, Kathmandu, Nepal, 6 Department of Internal Medicine, Section of Infectious Diseases, Yale School of Medicine, New Haven, Connecticut, United States of America

* roman.shrestha@yale.edu

## Abstract

Men who have sex with men (MSM) are disproportionately affected by HIV. Individuals who accessed sexual health clinic services for sexually transmitted infections (STIs) and pre-exposure prophylaxis (PrEP) were less likely to engage in high-risk sexual behaviors such as transactional sex and condomless sex and exhibited greater perceived importance of HIV testing. HIV testing is considered the gateway to both prevention and treatment of HIV, enabling timely intervention in HIV transmission. Therefore, this study aimed to measure and determine factors associated with never testing for HIV among MSM in Nepal. We conducted the population-based HIV bio-behavioral surveillance study between October and December 2022 using the respondent-driven sampling (RDS) method. We computed estimates for never testing HIV and conducted bivariate and multivariate analyses to explore the correlation between participant characteristics and never testing HIV. Among the 250 participants, over half of the participants (52.5%) had never tested for HIV in their lifetime, and only 11.7% had tested for HIV in the last 12 months. MSM who had not engaged in transactional sex (aOR:4.5; 95% CI: 1.2-17.3), had no daily internet access (aOR: 5.4; 95% CI: 1.4-21.3), had no prior diagnosis of sexually transmitted infection in their lifetime (aOR: 8.4; 95% CI: 2.8-25.2), had never heard of HIV self-testing (aOR:6.7; 95% CI: 2.8-16.0), and were unaware that someone taking PrEP (aOR:44.9; 95% CI: 10.5-191.6) had higher odds of never having been tested for HIV. Conversely, MSM who were single (aOR:0.3; 95% CI: 0.1-0.8) had lower odds of never being tested for HIV. This study highlights a significant gap in HIV testing among MSM in Nepal, particularly among those who were unaware of an HIV self-testing kit. The findings underscore the need for targeted interventions to address multi-level barriers to increase HIV testing rates among Nepali MSM.

**Data availability statement:** The raw data used for the analysis of this study have been deposited in the public data repository platform "Fig share" and can be easily accessed using the link 10.6084/m9.figshare.23929524.

**Funding:** We acknowledge financial support from a career development award from the National Institute on Drug Abuse (K01 DA051346) to Dr. Roman Shrestha. The funders had no role in study design, data collection, analysis, manuscript preparation, or the decision to publish.

**Competing interests:** The authors have declared that no competing interests exist.

## Introduction

Nearly 40 years after the first cases of HIV were reported, the number of people living with HIV is now estimated to be 40 million [1–3]. Men who have sex with men (MSM) are among the key populations disproportionately affected by HIV/AIDS [4,5], with MSM being at a 26 times higher risk of acquiring HIV than the general population [6]. HIV prevalence among MSM varies by region, with a median of 5% in Southeast Asia and 12.6% in Eastern and Southern Africa [6]. In 2019, MSM accounted for 44% of new HIV infections in Asia and the Pacific [6]. Globally, the HIV incidence rate among MSM has risen by 32% between 2010 and 2022 [7], which is deeply concerning given their existing stigma and marginalization, particularly in low and middle-income countries like Nepal [8,9]. In Nepal, the prevalence of HIV among MSM has increased from 3% in 2015 to 9% in 2018 [10,11]. However, only 2.5% of MSM reported their HIV infection, suggesting that a sizable portion (6.5%) may be unaware of their HIV status, have never tested for it, or are unwilling to disclose and seek treatment [10].

The Joint United Nations Program on HIV and AIDS (UNAIDS) has a global treatment goal of 95-95-95 targets by 2030 for epidemic control [12]. Nepal commits to this global goal as set out in the National HIV Strategic Plan 2021–2026 [11]. HIV testing is the entry point for HIV diagnosis, prevention, treatment, and care. It is essential for achieving the objectives of Nepal's National HIV Strategic Plan 2021–2026 of reducing new HIV infections, improving health outcomes of people living with HIV, and reducing HIV-related health inequalities among people living with HIV (PLHIV) and key populations [11]. Nepal's HIV testing and counseling services adhere to the 2022 National HIV Testing and Treatment Guidelines, offering pre-test information, post-test counseling, and linkage to prevention, care, and treatment services [13]. Facility-based, including voluntary counseling and testing, community-based, and self-testing services are available [13]. Key populations such as MSM and their partners in Nepal who test negative for HIV are required to retest every six months or sooner if they exhibit any symptoms suggestive of HIV [13]. This regular testing and awareness of one's HIV status are associated with risk reduction and behavior change, enabling newly diagnosed individuals to promptly connect to antiretroviral therapy (ART) care, and support services [14,15].

Despite the expansion of HIV testing services to 263 sites and the implementation of free testing schemes in most areas, HIV testing among MSM in Nepal remains stagnant [11]. There is a critical need to understand the underlying individual issues, barriers, and determinants that prevent Nepalese MSM from reaching and using HIV testing services. There is scant literature exploring the correlates of never testing for HIV, particularly among MSM in Nepal. Most existing research focuses on risk factors associated with HIV infections. Therefore, this study aims to identify factors associated with never testing for HIV among MSM in Nepal and to recommend targeted public health interventions to improve testing rates.

## Methods

### Ethics statement

The institutional review boards at the University of Connecticut (H22-0039) and the Nepal Health Research Council (Ref: 43; Protocol number: 239/2022 P) approved the study. Prior to initiating the survey, Research Assistants (RAs) read the consent form aloud and also provided an opportunity for participants to read the document themselves. Participants were clearly informed about the purpose of the study, the voluntary nature of their participation, and their right to withdraw at any time without facing any negative consequences. The RAs thoroughly explained the content of the consent form and addressed any questions or concerns raised by participants. Upon providing informed consent, participants digitally signed the form via Qualtrics and proceeded with the study procedures. All participants were required to provide written informed consent before commencing any study-related activities.

### Study design and participants

A population-based cross-sectional bio-behavioral respondent-driven survey was conducted from 1st October to 30th December 2022 in collaboration with the Blue Diamond Society (BDS), a community-based organization dedicated to improving the health and rights of sexual and gender minority populations. The study targeted MSM residing in the Kathmandu Valley, which includes Kathmandu, Bhaktapur, and Lalitpur districts. Individuals were eligible for the study if they identified themselves as cisgender MSM, were at least aged 18 years, had prior oral or anal sex with a male partner, and could read or understand either Nepali or English.

### Study procedure

The present study is part of a larger population-based HIV/AIDS bio-behavioral survey [16,17]. In this study, respondent-driven sampling (RDS) was used to recruit the participants. RDS is a chain-link sampling method that relies on social networks and is commonly used to access populations that are hard to reach [18,19]. The recruitment process began with identifying five MSM "seeds," purposefully selected based on recommendations from BDS. The selection of seeds ensured socio-demographic and geographic representation within the study sample.

Each seed participant who completed the interviewer-administered questionnaire received five recruitment coupons to distribute among their peers in the MSM community. Subsequent participants who received these coupons were encouraged to enroll their peers in the study, and each participant was given an additional set of five coupons to expand the recruitment chain further. All participants provided informed consent prior to engaging in any study-related activities, ensuring that ethical considerations were upheld throughout the research process.

Trained research assistants conducted face-to-face interviews with the participants using Qualtrics. The interviews were conducted in a private room to ensure privacy, and each session lasted approximately 40 minutes, allowing for a comprehensive exploration of the research topics. The participants received compensation of 1000 Nepalese Rupees (approximately USD 8). For each eligible peer successfully recruited into the study by a participant, an extra incentive of 500 Nepalese Rupees (approximately USD 4) was provided. These incentives served as compensation for transportation costs and any potential loss of work time incurred by the participants during their recruitment efforts (Fig 1).

### Study measures

The dependent variable (i.e., never testing for HIV) was measured using a single-item question, "Have you ever been tested for HIV?". The response was a dichotomous choice of "Yes" or "No".

In addition, participants were asked if they had undergone HIV testing in the last 12 months. For those who had been tested for HIV, the site of their last test was asked, including options such as a nongovernmental organization (NGO) or

PLOS Global Public Health

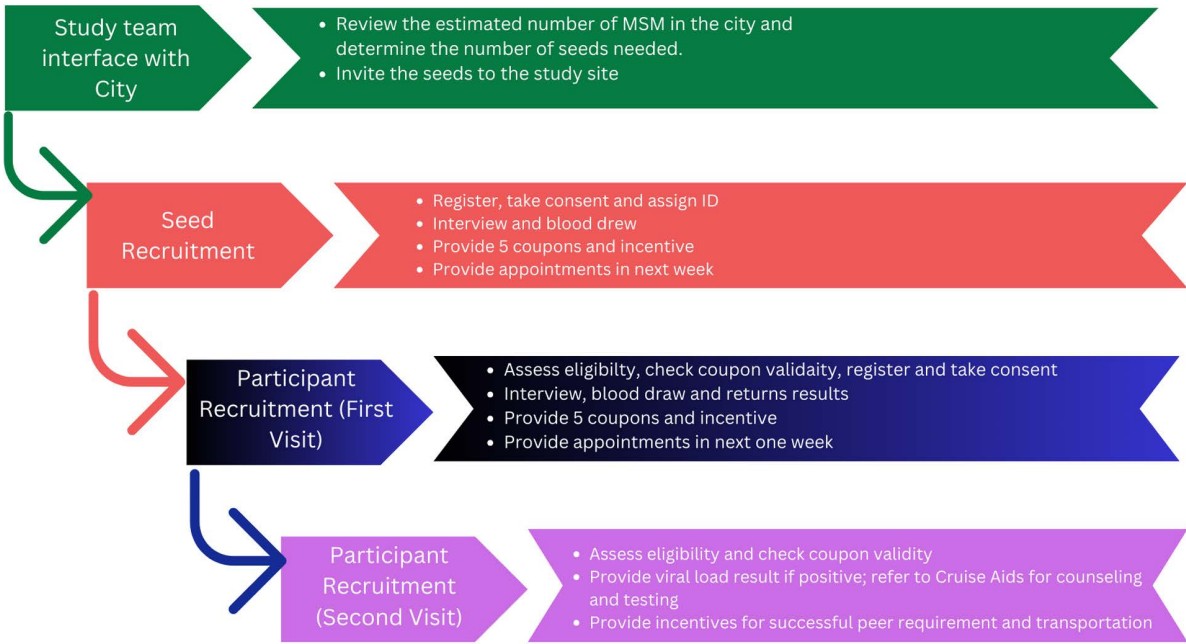

**Fig 1. Study procedure to recruit MSM from October to December 2022 in Kathmandu Valley, Nepal.**

community-based organization, government hospital, private clinic or hospital, and self-testing. Additionally, participants were asked about their awareness of HIV self-testing kits and their willingness to use them.

## Demographic characteristics

The sociodemographic information collected for this study included age, sexual orientation, educational status, and relationship status. Age was categorized into two groups: individuals younger than 25 and those 25 years old or older. Sexual orientation was assessed as either gay or bisexual. Educational levels were classified into less than higher secondary and higher secondary or above. Relationship status was categorized as either single or with a partner.

## HIV-related sexual risk behaviors

The assessment of HIV-related sexual risk behaviors included questions such as whether participants had ever engaged in transactional sex, had engaged in condomless sex within the last six months, and had ever been diagnosed with any sexually transmitted infection (STI). Additionally, participants were asked to report the number of sexual partners they had in the last six months, which was grouped as multiple sex partners (yes/no). Individuals who engaged in sexual activity with more than one partner were classified as having multiple sexual partners.

The participants' perceived MSM-related stigma was assessed using a 14-item scale based on the Neilands Sexual Stigma Scale, which was modified by Logie et al [20,21]. Responses to the 14 questions were measured on a 4-point Likert scale, scoring from 0 to 3, giving a total stigma score of 0–42. A score greater than 10 was considered stigmatized.

## Others

Participants were asked about their daily internet access and whether they knew individuals using pre-exposure prophylaxis (PrEP) or not.

The questionnaire used in this study is provided in the supplementary materials as Supplementary File (S1 File).

## Statistical analysis

The statistical software Stata.SE Corp version 17.0 was used for data analysis. We chose Stata because it has easy-to-use commands for logistic regression, a simple syntax, and a do-file system that ensures complete reproducibility. Descriptive statistics, including number and percentages for categorical variables and mean and standard deviation for continuous variables, were used to summarize the data. All the variables were presented in number and percentage form, except for age. We calculated the mean and standard deviation for age.

To minimize biases associated with chain referral sampling, weights were created in RDS analysis Tool 7.1 (RDSAT; Cornell, NY) using the RDSII estimator to account for the effect of differences in the social network sizes of participants. Weights were based on the transition matrix for the dependent variable (never tested for HIV). Network size was assessed using the response from the latter of two questions: "How many men who have had oral or anal sex with men in the last 12 months do you know, who also know you and live in this city?" and "among these men that you know personally, how many of them are 18 years and older?" RDSAT was used to calculate the weighted value. These weights were exported from RDSAT and merged into a Stata dataset comprising participants' demographic and behavioral data.

Bivariate logistic regression models were used to estimate the unadjusted association between the outcome variables: RDS-weighted prevalence and bootstrapped confidence intervals were calculated for all variables explored in regression modeling. Multivariate logistic regression models were built to estimate the adjusted association between never-testing HIV and the covariates. Bivariate and multivariate logistic regression models were also built with RDS weighting. P-values < 0.05 were used to indicate statistical significance.

## Results

### Participant's characteristics

The study recruited 250 MSM. The recruitment trees used in the study are shown in Fig 2. Table 1 presents the study's RDS-weighted sociodemographic characteristics. More than half of the participants, 54.1 (95% CI; 47.8-60.3), were less than 25 years old, and 50.9% (95% CI; 44.6-57.1) identified themselves as bisexual. About one-fourth of the study participants had multiple sex partners, and 71.4% (95% CI; 65.7-76.9) reported having condomless sex in the last six months.

### HIV testing and behavior

As shown in Table 2, more than half of the participants had never tested for HIV in their lifetime (52.5%: 95% CI; 46.3-58.8). Only 11.7% (95% CI; 7.7-15.7) had tested for HIV in the last 12 months, with the majority of them having tested for HIV at a community-based organization or NGO (81.8%: 95% CI;74.8-88.8). Only 37.9% (95% CI; 31.9-44.0) of the participants had heard about HIV self-testing, but after this study, 80.9% (95% CI; 75.9-85.8) showed interest in using HIV self-testing kits in the future.

### Multivariable logistic regression models for the predictors of never-tested HIV

Table 3 shows independent correlates of never testing for HIV in the multivariable logistic regression model. MSM who had not engaged in transactional sex (aOR:4.5; 95% CI: 1.2-17.3), had no daily internet access (aOR: 5.4; 95% CI: 1.4-21.3), had no prior diagnosis of STI (aOR: 8.4; 95% CI: 2.8-25.2), had never heard of HIV self-testing (aOR:6.7; 95% CI: 2.8-16.0), and were unaware that someone taking PrEP (aOR:44.9; 95% CI: 10.5-191.6) had higher odds of never being tested for HIV. MSM who were single (aOR:0.3; 95% CI: 0.1-0.8) had lower odds of never being tested for HIV.

## Discussion

Although the government of Nepal provides free-of-charge voluntary HIV counseling and testing, the proportion of MSM who have never tested for HIV was discouragingly high in our sample. WHO guidelines recommend annual retesting

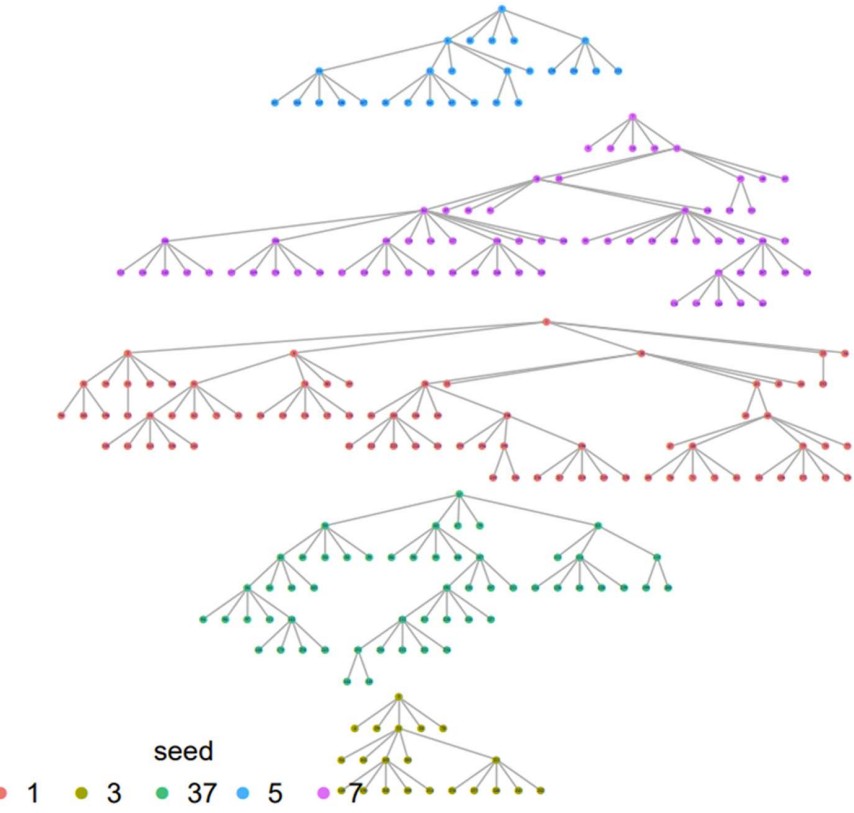

**Fig 2. Respondent-driven sampling network diagram of the seeds and waves.**

for MSM and more frequent retesting (every 3–6 months) for those with risk factors [22]. Despite engagement in risk behaviors (e.g., condomless sex) in our sample, retesting was relatively low, which aligns with findings from other studies [23,24]. The prevalence of multiple sexual partners, involvement in transactional sex, and instances of condomless intercourse over the past six months is notably high. These risky sexual behaviors could significantly lead to the rapid transmission of HIV within sexual networks. The majority of MSM tested for HIV in NGOs or community organizations, which is unsurprising given the stigma associated with traditional healthcare settings for HIV testing in Nepal. This might be because of a lack of awareness of HIV testing, concern about HIV testing confidentiality, negative emotions such as the inconvenience of detection, and anxiety about HIV-related stigma. Nearly half of the participants were unfamiliar with HIV self-testing. These findings highlight the need for enhanced research and programmatic to promote regular HIV testing among MSM and ensure that those diagnosed are linked to integrated care services. HIV self-testing has emerged as a promising strategy to supplement standard facility-based testing efforts since it is a widely available and flexible HIV testing option, overcoming the barriers of stigmatization and privacy [25]. Our findings, which indicated a high willingness to use HIV self-testing in our sample, highlight the opportunity for HIV self-testing to complement traditional testing programs by removing barriers and increasing access to HIV testing for key populations.

Our results revealed that single individuals were less likely to be tested for HIV when compared to individuals in relationships. MSM in relationships often underestimate their HIV risk and engage in condomless anal sex as a way to show love, intimacy, and trust towards each other in strengthening their relationship commitment [26]. This emotional bond with their partner may also act as a barrier to routine HIV testing, as it could be seen as a sign of distrust in the relationship.

**Table 1. General characteristics of the participants.**

| Characteristics | Number | Crude % | RDS weighted %, (95% CI) |
|---|---|---|---|
| **Socio-demographic characteristics** | | | |
| Age groups (years) (Mean ±SD = 27.6 ± 8.9) | | | |
| Less than 25 | 127 | 50.8 | 54.1 (47.8–60.3) |
| 25 and above | 123 | 49.2 | 45.9 (39.7–52.1) |
| Sexual orientation | | | |
| Gay | 158 | 63.2 | 49.1 (42.9–55.4) |
| Bisexual | 92 | 36.8 | 50.9 (44.6–57.1) |
| Educational status | | | |
| Less than higher secondary | 105 | 42.0 | 47.4 (41.2–53.6) |
| Higher secondary and above | 145 | 58.0 | 52.6 (46.4–58.8) |
| Relationship status | | | |
| Single | 161 | 64.4 | 67.2 (61.4–73.1) |
| With Partner | 89 | 35.6 | 32.8 (26.9–38.6) |
| **HIV-related and sexual risk behaviors** | | | |
| Multiple sex partners in the last six months | | | |
| Yes | 101 | 40.4 | 26.9 (21.4–32.4) |
| No | 149 | 59.6 | 73.1 (67.6–78.6) |
| Ever engaged in transactional sex | | | |
| Yes | 55 | 22.0 | 13.7 (9.4–17.9) |
| No | 195 | 78.0 | 86.3 (82.1–90.6) |
| Condomless sex in the last six months | | | |
| Yes | 156 | 62.4 | 71.4 (65.7–76.9) |
| No | 94 | 37.6 | 28.6 (23.0–34.3) |
| Ever diagnosed with any STIs | | | |
| Yes | 57 | 22.8 | 17.7 (13.0–22.5) |
| No | 193 | 77.2 | 82.2 (77.5–87.0) |
| MSM related stigma | | | |
| Yes | 61 | 24.4 | 13.9 (9.6–18.2) |
| No | 189 | 75.6 | 86.1 (81.8–90.4) |
| **Others** | | | |
| Daily internet access | | | |
| Yes | 219 | 87.6 | 86.8 (826–91.0) |
| No | 31 | 12.4 | 13.2 (9.0–17.4) |
| Know someone taking PrEP | | | |
| Yes | 101 | 40.4 | 25.1 (19.7–30.5) |
| No | 149 | 59.6 | 74.9 (69.5–80.3) |

findings align with previous studies that reported low rates of discussing HIV testing or serostatus among MSM in relationships, often relying on trust rather than safer sex practices [24,27]. Therefore, to address low testing rates among MSM in a relationship, interventions should focus on couples HIV testing and counseling, as well as utilizing online platforms to reach MSM.

MSM who never engaged in transactional sex had higher odds of never being tested for HIV, which is consistent with findings from previous studies [28,29]. MSM involved in transactional sex were more likely to engage in condomless anal intercourse and substance use during sex, increasing their HIV risk [29]. They also showed higher rates of sexually

**Table 2. Characteristics of HIV testing and behavior of the participants.**

| Characteristics | Number | Crude % | RDS weighted %, (95% CI) |
|---|---|---|---|
| **HIV testing and behavior** | | | |
| Ever tested HIV | | | |
| Yes | 165 | 66.0 | 47.5 (41.3–53.7) |
| No | 85 | 34.0 | 52.5 (46.3–58.8) |
| Tested HIV in the last 12 months | | | |
| Yes | 40 | 16.0 | 11.7 (7.7–15.7) |
| No | 210 | 84.0 | 88.3 (84.3–92.3) |
| Site of last HIV testing (n = 165) | | | |
| NGO or community-based organization | 134 | 81.2 | 81.8 (74.8–88.8) |
| Government hospital | 15 | 9.1 | 6.6 (2.1–11.1) |
| Private clinic or hospital | 11 | 6.7 | 7.1 (2.5–11.8) |
| Self-test | 5 | 3.0 | 4.5 (0.7–8.3) |
| Heard of HIV self-testing | | | |
| Yes | 132 | 52.8 | 37.9 (31.9–44.0) |
| No | 118 | 47.2 | 62.1 (56.0–68.1) |
| Willingness to use HIV self-testing | | | |
| Yes | 206 | 82.4 | 80.9 (75.9–85.8) |
| No | 44 | 17.6 | 19.1 (14.2–24.0) |

transmitted infections (STIs) and group sex participation [30]. As in our study, MSM who did not have a prior history of STI had higher odds of never testing HIV. MSM diagnosed with STIs may have undergone HIV testing concurrent with previous STI diagnoses [31]. MSM may have been aware of their elevated HIV risk and engaged in HIV testing to maintain their health. In Kathmandu, some organizations work for MSM and sex workers on HIV prevention outreach, from which MSM might have benefitted. Prior data have shown that STIs are both a marker of high-risk behavior and a possible factor in HIV acquisition [31,32]. Healthcare workers and service providers can offer HIV or STI testing, assess sexual health history at the same time during the clinic's visit, and provide suggestions for HIV prevention.

Individuals who have heard of HIV self-testing and know someone taking PrEP are more likely to undergo HIV testing, potentially due to increased awareness of HIV prevention strategies, normalization of testing practices, and perceived accessibility and convenience of testing options. This is supported by other similar study findings, where individuals never heard of HIV- self-testing and PrEP had higher odds of never testing for HIV [5,25,33]. The finding that the participants who do not know of anyone taking PrEP are very highly likely to never test for HIV suggests a significant association between social awareness of PrEP and HIV testing behavior. HIV self-testing combined with PrEP information can be extended to individuals lacking knowledge of PrEP or who have been tested for HIV through PrEP clients' social networks. HIV testing is an essential entry point for HIV prevention services, such as PrEP [34]. Not knowing any acquaintances using PrEP might serve to less perceived relevance or urgency about HIV testing, leading to a more passive approach towards HIV prevention. Therefore, increasing visibility and discussions about PrEP within social networks might play a crucial role in encouraging more MSM to engage in regular HIV testing.

Participants who did not have access to daily internet were shown to have higher rates of never testing for HIV. This is because there is a wide range of health-related activities for HIV on the internet [35]. People who do not have access to the daily internet might be deprived of information about HIV, including the importance of testing, available testing approaches, and service areas. The Internet can help individuals find local testing centers, clinics, and other resources, deprivation of which might lead MSM never to check their HIV status. In addition, MSM with daily internet access might

**Table 3. Factors associated with never testing HIV among MSM in Nepal (N = 250).**

| Characteristics | Crude weighted OR (95% CI) | Adjusted weighted OR (95% CI) |
|---|---|---|
| Sexual orientation | | |
| Gay | 1 [Reference] | 1 [Reference] |
| Bisexual | 3.6 (2.1–6.2) | 1.1 (0.5–2.5) |
| Educational status | | |
| Less than higher secondary | 1 [Reference] | 1 [Reference] |
| Higher secondary and above | 0.4 (0.3–0.7) | 0.9 (0.4–1.9) |
| Relationship status | | |
| With partner | 1 [Reference] | 1 [Reference] |
| Single | 0.5 (0.3–0.8) | 0.3 (0.1–0.8) |
| MSM related stigma | | |
| Yes | 1 [Reference] | 1 [Reference] |
| No | 13.6 (4.3–43.1) | 3.3 (0.8–13.7) |
| Multiple sex partners in the last six months | | |
| Yes | 1 [Reference] | 1 [Reference] |
| No | 1.7 (0.9–2.9) | 0.4 (0.1–1.1) |
| Ever engaged in transactional sex | | |
| Yes | 1 [Reference] | 1 [Reference] |
| No | 2.1 (0.9–4.4) | 4.5 (1.2–17.3) |
| Daily internet access | | |
| Yes | 1 [Reference] | 1 [Reference] |
| No | 2.9 (1.3–6.5) | 5.4 (1.4–21.3) |
| Heard of HIV self-testing | | |
| Yes | 1 [Reference] | 1 [Reference] |
| No | 8.2 (4.5–14.8) | 6.7 (2.8–16.0) |
| Ever diagnosed with any STIs | | |
| Yes | 1 [Reference] | 1 [Reference] |
| No | 5.7 (2.6–12.3) | 8.4 (2.8–25.2) |
| Know of someone taking PrEP | | |
| Yes | 1 [Reference] | 1 [Reference] |
| No | 52.8 (14.4–193.7) | 44.9 (10.5–191.6) |

be active in geo social networking applications (GSN) like Grindr, Hornet, and Jack'd. There have been several types of interventions for HIV and STI awareness and testing on that app, which could be a motivating factor for MSM to undergo HIV testing [36]. Since there is a high willingness to use an HIV self-testing kit, HIV self-testing using the internet can be replicated in Nepal, which has already proven feasible and acceptable in increasing HIV testing uptake among MSM in other countries [36–38]. Tailoring digital health technologies such as social media, text messaging, mHealth, and eHealth platforms can drive demand for HIV testing services among MSM by making them aware and ultimately improving testing intentions. Digital health interventions via GSN applications, combined with social marketing and peer outreach, could be effective strategies to promote HIV testing and early linkage to HIV treatment among MSM who face barriers like anti-LBGT legislation and healthcare discrimination of MSM and HIV in healthcare settings.

This study was the first to investigate the prevalence and factors associated with never testing HIV, focusing on MSM in Nepal by using the RDS method. A review of more than 120 RDS studies conducted worldwide found that RDS is an effective technique when designed and implemented appropriately for sampling most at-risk populations

for HIV biological and behavioral surveys [39]. Despite its strength, this study has some limitations that must be acknowledged. First, monetary incentives for participation in RDS may have appealed more strongly to lower socio-economic status MSM than higher. Second, our study was conducted in the Kathmandu Valley. Thus, the findings cannot be generalizable to the overall MSM in Nepal. Third, we did not utilize a standardized questionnaire to assess participants' knowledge of HIV testing or the accessibility and availability of HIV testing programs. Fourth, the face-to-face interview format may have influenced participant's responses due to social desirability bias. Fifth, the use of RDS may have introduced selection biases and limited the generalizability of the findings. Finally, the sample size was modest. Therefore, some of the nonsignificant associations observed could have occurred because of inadequate power.

## Conclusion

Our study showed that more than half of the participants were never tested for HIV. To increase HIV testing, it is imperative to reach out to people who are unaware of an HIV self-testing kit. Digital health and traditional interventions should be integrated with HIV prevention programs to engage key populations in accessing HIV testing. HIV testing requires more promotion that focuses on at-risk MSM who are involved in risky sexual behavior, single, and do not have access to the internet daily. Moreover, efforts should be made to reduce HIV-related stigma and strengthen the training of healthcare providers and peer educators to improve the dissemination of HIV knowledge.

## Supporting information

**S1 File. Questionnaire.**
(PDF)

**S2 File. Checklist.**
(DOCX)

## Author contributions

**Conceptualization:** Kiran Paudel, Roman Shrestha.

**Data curation:** Kiran Paudel, Kamal Gautam, Roman Shrestha.

**Formal analysis:** Kiran Paudel, Roman Shrestha.

**Funding acquisition:** Manisha Dhakal, Roman Shrestha.

**Investigation:** Manisha Dhakal, Roman Shrestha.

**Methodology:** Kiran Paudel, Kamal Gautam, Anjila Pandey, Prashamsa Bhandari, Md. Safaet Hossain Sujan, Jefferey A Wickersham, Roman Shrestha.

**Project administration:** Kiran Paudel, Kamal Gautam, Anjila Pandey, Manisha Dhakal, Md. Safaet Hossain Sujan, Jefferey A Wickersham, Roman Shrestha.

**Resources:** Kamal Gautam, Manisha Dhakal.

**Software:** Kiran Paudel, Kamal Gautam.

**Supervision:** Jefferey A Wickersham, Roman Shrestha.

**Writing – original draft:** Kiran Paudel, Kamal Gautam, Anjila Pandey, Prashamsa Bhandari, Md. Safaet Hossain Sujan, Jefferey A Wickersham, Roman Shrestha.

**Writing – review & editing:** Kiran Paudel, Kamal Gautam, Anjila Pandey, Prashamsa Bhandari, Jefferey A Wickersham, Roman Shrestha.

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
