## [Decision Letter · Decision Letter 0]

20 Feb 2025

PGPH-D-24-02775

Correlates of never testing for HIV among men who have sex with men in Nepal

Dear Dr. Shrestha,

Thank you for submitting your manuscript to PLOS Global Public Health. After careful consideration, we feel that it has merit but does not fully meet PLOS Global Public Health’s publication criteria as it currently stands. Therefore, we invite you to submit a revised version of the manuscript that addresses the points raised during the review process.

The reviewers noted that the manuscript provides important insights in addressing barriers of HIV testing. However, some clarifications should be made to the Methods and Discussions sections before it can be accepted for publication.

We look forward to receiving your revised manuscript.

Kind regards,

Tsz Ho Kwan

Academic Editor

Journal Requirements:

1.We noticed you have some minor occurrence of overlapping text with the following previous publication(s), which needs to be addressed:

https://nature.com/articles/s41598-023-27485-8

In your revision ensure you cite all your sources (including your own works), and quote or rephrase any duplicated text outside the methods section. Further consideration is dependent on these concerns being addressed.

Additional Editor Comments (if provided):

Reviewers' comments:

Reviewer's Responses to Questions

**Comments to the Author**

1. Does this manuscript meet PLOS Global Public Health’s publication criteria ? Is the manuscript technically sound, and do the data support the conclusions? The manuscript must describe methodologically and ethically rigorous research with conclusions that are appropriately drawn based on the data presented.

Reviewer #1: Yes

Reviewer #2: Yes

2. Has the statistical analysis been performed appropriately and rigorously?

Reviewer #1: Yes

Reviewer #2: Yes

3. Have the authors made all data underlying the findings in their manuscript fully available (please refer to the Data Availability Statement at the start of the manuscript PDF file)?

Reviewer #1: No

Reviewer #2: Yes

4. Is the manuscript presented in an intelligible fashion and written in standard English?

Reviewer #1: Yes

Reviewer #2: Yes

5. Review Comments to the Author

Reviewer #1: The manuscript is sound and the conclusion is key to increase knowledge in the field of study. Due to the nature of the variables, the bivariate and multivariate analyses utilized seem appropriate to the study. There are however, some observations;

Abstract

The Information in the Abstract doesn't sufficiently highlight MSM engaging in transactional sex. If the findings among this group is the key outcome of the study, then it should clearly be highlighted the Abstract.

What duration were they diagnosed with STIs? Is it a recent diagnosis? Will be great to qualify the "diagnosed with STIs" as "recently diagnosed" to enable adequate classification of the risk of acquiring the virus.

What percentage haven't tested due to Knowledge of testing using other kits, availability of kits in Health facilities, Community or Private sector, availability and accessibility to testing programs?

Main body

45; Why the focus on only Structural barriers? Is there specific role of Behavioral and Biomedical influence on the dependent variables? The statement of objective in the Abstract section doesn’t specify the structural factors as the key independent variables.

50; If the focus is mainly on the influence of structural factors, then it should be stated in the main objective.

76; Any Global HIV Prevalence data among MSM? Would have been great to include this.

79; It Would be nice to include data on HIV Prevalence among MSM in the region and/or Globally.

163; What was the rationale for the 25yr mark? Is it in line with the cut off for the young persons” population In the Country.

245; It Would be great to compare the literature findings with the outcomes in the discussion section.

Reviewer #2: The manuscript presents an important analysis of HIV testing behaviors among men who have sex with men (MSM) in Nepal. The use of respondent-driven sampling (RDS) and multivariate analysis provides some insights into factors associated with never testing for HIV. The findings have the potential to inform targeted interventions aimed at increasing HIV testing among MSM in Nepal. However, there are several areas that require improvement, including clarity in the presentation of results, citation accuracy, methodological justifications, and addressing study limitations.

• Line 76 – The citation is inaccurate. The referenced source discusses Asia and the Pacific region, but the manuscript refers to global trends. Please correct this or provide a more appropriate global reference.

• Line 73 – The authors should cite the latest Global Burden of Disease (GBD) study and data for accuracy. A suitable reference is: https://www.thelancet.com/journals/lanhiv/article/PIIS2352-3018(24)00212-1/fulltext.

• Line 329 – The manuscript lacks a citation for the RDS methodology. A standard reference for RDS should be included.

• Line 163 – The age categorization (<25 vs. ≥25) needs justification. Why was this cutoff chosen? Would alternative classifications (e.g., using smaller intervals) provide more different results?

• Line 173 – The manuscript states that participants were "grouped as multiple sex partners (yes/no)," but does not define the threshold for "multiple." The exact number of partners considered "yes" should be clarified.

• Line 336-338 – The manuscript does not explain why the sample size was not increased. Was the sample size pre-determined?

• Line 230-232 – The manuscript can include confidence intervals in the text when discussing findings.

• Line 274 – The findings in this paragraph are unclear. The discussion focuses on previous studies, but there is no clear linkage to the study's own results.

• Line 272 – The phrase "This trust and emotional connection with their partner can also act as a barrier to regular testing like..." is incorrectly structured and should be rephrased for clarity.

• Line 333 – The limitations section can address other limitations:

o Due to the study’s cross-sectional design, causal relationships cannot be established.

o Potential limitations of RDS method.

o The face-to-face interview method may have influenced participants’ responses.

• Line 98 – The abbreviation ART is not defined upon first use.

• Line 117-121 – The phrasing is repetitive and redundant.

6. PLOS authors have the option to publish the peer review history of their article (what does this mean? ). If published, this will include your full peer review and any attached files.

**Do you want your identity to be public for this peer review?** For information about this choice, including consent withdrawal, please see our Privacy Policy .

Reviewer #1: **Yes: ** Dr, Joseph Teryima Ashivor

Reviewer #2: No

---

## [Decision Letter · Decision Letter 1]

1 Apr 2025

PGPH-D-24-02775R1

Correlates of never testing for HIV among men who have sex with men in Nepal

Dear Dr. Shrestha,

Thank you for submitting your manuscript to PLOS Global Public Health. After careful consideration, we feel that it has merit but does not fully meet PLOS Global Public Health’s publication criteria as it currently stands. Therefore, we invite you to submit a revised version of the manuscript that addresses the points raised during the review process.

There are minor remarks to be addressed, as raised by the reviewer. In addition, please report the odds ratios to two decimal places.

We look forward to receiving your revised manuscript.

Kind regards,

Tsz Ho Kwan

Academic Editor

Journal Requirements:

Additional Editor Comments (if provided):

Reviewers' comments:

Reviewer's Responses to Questions

**Comments to the Author**

1. If the authors have adequately addressed your comments raised in a previous round of review and you feel that this manuscript is now acceptable for publication, you may indicate that here to bypass the “Comments to the Author” section, enter your conflict of interest statement in the “Confidential to Editor” section, and submit your "Accept" recommendation.

Reviewer #1: All comments have been addressed

2. Does this manuscript meet PLOS Global Public Health’s publication criteria ? Is the manuscript technically sound, and do the data support the conclusions? The manuscript must describe methodologically and ethically rigorous research with conclusions that are appropriately drawn based on the data presented.

Reviewer #1: Yes

3. Has the statistical analysis been performed appropriately and rigorously?

Reviewer #1: Yes

4. Have the authors made all data underlying the findings in their manuscript fully available (please refer to the Data Availability Statement at the start of the manuscript PDF file)?

Reviewer #1: Yes

5. Is the manuscript presented in an intelligible fashion and written in standard English?

Reviewer #1: Yes

6. Review Comments to the Author

Reviewer #1: ABSTRACT

The abstract addresses the barriers to HIV testing, which is crucial for both HIV prevention and treatment.

• The aim of the study is clearly stated, and the focus on understanding the factors related to never having tested for HIV among MSM is both timely and important.

• The design is clearly stated, leveraging on a respondent-driven sampling (RDS), a robust method for sampling hidden populations like MSM, which enhances the generalizability of the findings to the broader MSM population in Nepal. The use of both bivariate and multivariate analyses is appropriate for understanding the relationships between participant characteristics and HIV testing.

• The abstract clearly states statistically significant findings (e.g., aORs and 95% CI) that identify factors associated with never testing for HIV, such as lack of awareness about HIV self-testing and PrEP, lack of engagement in transactional sex, and absence of daily internet access.

INTRODUCTION

• The introduction effectively situates the research within the broader global context of HIV and specifically highlights the vulnerable position of MSM. It is clear and informative in terms of the scale of the HIV epidemic.

• The introduction brings attention to the specific context of Nepal, which is crucial for understanding the regional challenges and the need for focused interventions.

• The rationale for the study, including the gap in the literature and the critical need for exploring the correlates of "never testing," is clearly articulated.

However, the final sentence of the introduction can be refined to reflect;

110. "Therefore, this study aims to identify factors associated with never testing for HIV among MSM in Nepal and to recommend targeted public health interventions to improve testing rates."

METHODOLOGY

The methodology is robust and well-designed, using established techniques like RDS to reach the MSM population. Previous recommendations have been incorporated.

STATISTICAL ANALYSIS

191; It might be useful to briefly clarify why the Stata.SE Corp version 17.0 and RDSAT (RDS Analysis Tool 7.1) is chosen for the study.

193; It will be great to specify the continuous and categorical variable used for the descriptive analysis.

RESULTS

The Results section is well-structured and clear. The findings are presented logically with appropriate statistical backing, and the tables help summarize the data effectively.

However, it is recommended that the authors present a brief introductory sentence for the Results section and ensure consistency in terminology.

DISCUSSION

This Discussion section is well-written and provides a thorough interpretation of the findings. The section also incorporates the previous observations.

Consider linking the limitations of the study on the potential impact of the findings.

CONCLUSION

The conclusion is clear and summarizes the main findings effectively, while incorporating previous observations.

7. PLOS authors have the option to publish the peer review history of their article (what does this mean? ). If published, this will include your full peer review and any attached files.

**Do you want your identity to be public for this peer review?** For information about this choice, including consent withdrawal, please see our Privacy Policy .

Reviewer #1: **Yes: ** Ashivor, Joseph .T.

---

## [Editor Report · Decision Letter 2]

14 Apr 2025

Correlates of never testing for HIV among men who have sex with men in Nepal

PGPH-D-24-02775R2

Dear Dr. Shrestha,

We are pleased to inform you that your manuscript 'Correlates of never testing for HIV among men who have sex with men in Nepal' has been provisionally accepted for publication in PLOS Global Public Health.

Best regards,

Tsz Ho Kwan

Academic Editor